# Magneto-Luminescent Nanocomposites Based on Carbon Dots and Ferrite with Potential for Bioapplication

**DOI:** 10.3390/nano12091396

**Published:** 2022-04-19

**Authors:** Mariia Stepanova, Aliaksei Dubavik, Arina Efimova, Mariya Konovalova, Elena Svirshchevskaya, Viktor Zakharov, Anna Orlova

**Affiliations:** 1International Laboratory Hybrid Nanostructures for Biomedicine, ITMO University, Saint Petersburg 199034, Russia; adubavik@itmo.ru (A.D.); melpomennia@gmail.com (A.E.); vvzakharov@itmo.ru (V.Z.); a.o.orlova@itmo.ru (A.O.); 2Department of Immunology, Shemyakin-Ovchinnikov Institute of Bioorganic Chemistry RAS, Moscow 117997, Russia; mariya.v.konovalova@gmail.com (M.K.); esvir@yandex.ru (E.S.)

**Keywords:** magneto-luminescent nanocomposites, photoluminescence, absorption, photostability, cytotoxicity, confocal cell visualization

## Abstract

Multifunctional nanocomposites that combine both magnetic and photoluminescent (PL) properties provide significant advantages for nanomedical applications. In this work, a one-stage synthesis of magneto-luminescent nanocomposites (MLNC) with subsequent stabilization is proposed. Microwave synthesis of magnetic carbon dots (M-CDs) was carried out using precursors of carbon dots and magnetic nanoparticles. The effect of stabilization on the morphological and optical properties of nanocomposites has been evaluated. Both types of nanocomposites demonstrate magnetic and PL properties simultaneously. The resulting MLNCs demonstrated excellent solubility in water, tunable PL with a quantum yield of up to 28%, high photostability, and good cytocompatibility. Meanwhile, confocal fluorescence imaging showed that M-CDs were localized in the cell nuclei. Consequently, the multifunctional nanocomposites M-CDs are promising candidates for bioimaging and therapy.

## 1. Introduction

Magneto-luminescent nanocomposites (MLNC) represent a new class of multifunctional nanomaterials combining magnetic and fluorescent signals in one sample. Nowadays, MLNCs are recognized as promising candidates in nanomedicine which can be used for diagnostic purposes, such as magnetic resonance imaging, targeted drug delivery, hyperthermic cancer therapy, as well as for some other applications [1,2,3,4,5,6,7,8,9,10].

There are various approaches to produce MLNCs, which can be split into several main groups according to the number of stages in the synthesis process. The first group of MLNCs synthesis represents a three-stage method. A number of articles are devoted to the description of various techniques to separately synthesize magnetic and luminescent particles for their subsequent binding [1]. In this case, the binding mechanisms vary widely by conjugating a luminescent particle onto a magnetic surface and vice versa [11,12,13,14,15]. Organic dyes, quantum dots, or specific polymers are used as phosphors [16,17,18]. 

The next group presents a two-stage approach, which includes methods for the production of a hybrid material, in which fluorescent particles are synthesized in the presence of ready-made magnetic nanoparticles (NPs). The stability of the nanocomposites obtained by this technique is achieved by creating a fluorescent shell on the surface of magnetic NPs [19,20]. In this case, the size of the final hybrid structures depends on the size of the initial magnetic particles. Despite the dependence of nanocomposite size on the size of the magnetic core, this synthesis technique leads to a higher polydispersity, resulting in a poorly controlled ratio of the magnetic and fluorescent components, since these parameters vary from particle to particle [21,22,23]. Moreover, magnetic nanoparticle surface passivation by a fluorescent shell leads to a weakening of nanocomposites’ magnetic characteristics. At the same time, the luminescence is quenched as a result of the efficient energy/charge transfer from the luminophore to the magnetic part. Thus, when two different materials are combined, a weakening of both the magnetic and luminescent properties of the nanocomposite occurs.

The third group includes one-stage synthesis, in which MLNCs can be obtained by the simultaneous synthesis of magnetic and fluorescent parts in one vessel. This approach allows immediate generation of a hybrid material, which can later be modified [24]. MLNCs can be obtained by hydrothermal or solvothermal bottom-up synthesis from precursors of magnetic and fluorescent NPs [25]. Comparing one-stage synthesis of MLNCs with others, one can highlight such advantages as synthesis simplicity and nanocomposite small size. In order to obtain high-quality chemically stable MLNCs, it is important to use precursors of proper quality.

Precursors of superparamagnetic iron oxide (Fe_3_O_4_) NPs (MNPs) can be used to obtain MLNCs since these NPs demonstrate good biocompatibility and excellent magnetic properties [26]. Despite the fact that many syntheses methods have now been proposed, the issue of the colloidal stability of MNPs has not yet been resolved, which is undoubtedly an important factor for further biomedical applications. For this reason, many works are aimed at improving the chemical stability of MNPs. Modification of the Fe_3_O_4_ surface with different materials is one of the ways to improve the colloidal stability of NPs. In general, these materials include two categories: organic (polyethylene glycol [27], polyaniline [28], chitosan [29,30]) and inorganic (silicon dioxide [31], Au [32,33], Ag [34], and carbon [35]) ones. 

Carbon dots (CDs) represent a class of fluorescent carbon-based NPs. They are promising as a luminescent part of MLNCs because their structure consists of an sp^2^-hybridization carbon matrix that stabilizes MNPs [36,37]. Taking advantage of the electrostatic repulsion generated by CDs, they can provide excellent colloidal stability to MLNCs. At the same time, CDs have such attractive properties as the feasibility of the synthesis, low toxicity, large surface area, good colloidal stability, the possibility of surface functionalization, high photoluminescence quantum yield, and excitation-dependent emission, and photostability [38]. Thus, due to the simplicity of the one-step simultaneous synthesis, high biocompatibility and strong magnetic properties of MNPs in a combination with the attractive fluorescent properties of CDs, they can find a wide application in the development and preparation of MLNCs. 

Here, a one-step approach to obtaining MLNC was proposed, in which the precursors for CDs and Fe_3_O_4_ NPs were used. Using this approach, we synthesized magnetic luminescent carbon dots. A carbon matrix with luminophores was formed during the synthesis and the size of the final nanocomposites was about 60 nm, which is an important parameter, for example, to penetrate cell nuclei.

The main advantage of such nanocomposites in biological applications is considered to be the absence of toxic materials and, therefore, high biocompatibility. However, the problem of combining crystal lattices of different materials remains. In this regard, post-synthesis treatment of M-CDs with glutathione was used to increase the stability of the obtained MLNCs. In addition, stabilization of M-CDs increases the resistance to destruction in living tissues, as well as to the influence of other external factors. Moreover, it prevents NPs aggregation, which is typical for MLNCs obtained by one-step synthesis from CDs and MNP precursors [24]. Thus, the proposed one-step method of MLNCs synthesis with subsequent stabilization has potential for biological applications, since MLNCs have a small size, exhibit high colloidal and decomposition stability, magnetic receptivity, and effective photoluminescence (PL).

## 2. Materials and Methods

### 2.1. Materials

The following reagents were used for the synthesis of CDs: citric acid (99.5%), ethylenediamine (≥99.5%), iron (II) chloride tetrahydrate, L-glutathione, MTT (all from Merck KGaA, Darmstadt, Germany) were used to stabilize the particles. Dimethyl sulfoxide (DMSO), paraformaldehyde (PFA) (Khimmed, Moscow, Russia). Deionized water (Milli-Q) was used as a solvent and in the experiments.

### 2.2. Synthesis of Magneto-Luminescent Carbon Dots

Magneto-luminescent carbon dots M-CDs were synthesized using an autoclaving/microwaving system (Milestone, Sorisole, Italy). Citric acid (2 g), ethylenediamine (1 mL) and iron (II) chloride tetrahydrate (FeCl_2_·4H_2_O) (0.6 g) were dissolved in deionized water until a clear solution was formed. After that, the solution was placed in the autoclave and heated at a 180 °C temperature and 25 bar pressure (700 W microwave power) for 3 min. After heat treatment, the reaction product was cooled to room temperature and centrifuged at 8000 rpm for 5 min. Supernatant fluid containing smaller particles was collected. The concentration of the stock solution was 62 mg/mL. The prepared stock M-CDs can be dried in an oven and redissolved in water for further use.

### 2.3. Stabilization of Magneto-Luminescent Carbon Dots

Stabilization of 8 mL M-CDs stock solution was carried out by post-synthesis treatment. The stabilizing agent L-glutathione (0.96 g) was added to the stock solution by stirring for 30 min. After that, the resulting particles were centrifuged at 4000 rpm for 5 min and then magnetized. After that, the color of the solution became darker while the PL of the solution under UV radiation was preserved. The concentration of the stabilized particles was 162 mg/mL. The prepared stabilized M-CDs (sM-CDs) can be dried in an oven and redissolved in water for further use.

### 2.4. Sample Characterization Methods

Scanning transmission electron microscopy (STEM) images and energy-dispersive X-ray spectroscopy (EDX) atomic percent data were obtained using a scanning electron microscope (Merlin, Ottobrunn, Germany). Dry MLNCs size was estimated from SEM images using the ImageJ program. The collected statistics of particle size distribution were processed by the OriginLab Program (Northampton, USA). The average size was set by a Gaussian approximation. Size distribution in water and ζ-potential were measured at 25 °C by dynamic light scattering (DLS) using a Zetasizer Nano ZS instrument (Malvern Instruments Ltd. Worcestershire, UK). The absorption spectra of the samples were recorded on a UV Probe 3600 spectrophotometer (Shimadzu, Kyoto, Japan) in the wavelength range of 300–800 nm at a monochromator slit width of 8 nm using an integrating sphere. The PL and PL excitation spectra were recorded using a Cary Eclipse spectrofluorimeter (Varian, Belrose, Australia) with a slit width of the exciting and recording elements of 5 nm. The PL lifetime was measured by a time-correlated single-photon counting technique using a MicroTime-100 confocal laser scanning microscope (PicoQuant, Berlin, Germany) at a 405 nm excitation wavelength. The average lifetime was calculated by exponential approximation. The amplitude-weighted average PL lifetime was determined by the following Equation (1):(1)$$\tau_{av}=\frac{A_{1}\tau_{1}^{2}+A_{2}\tau_{2}^{2}}{A_{1}\tau_{1}+A_{2}\tau_{2}}$$
where *A_i_* denoted fractional weights of various decay time components *τ_i_* of the multiexponential fitting. The PL quantum yield (QY) of M-CDs was defined as the relative quantum yield. Rhodamine 6G (with a standard QY of 95% in ethanol at 350 nm excitation) was used as a reference. The PL QY was determined using the following Equation (2):(2)$$\Phi_{S}=\Phi_{R}\times \frac{I_{S}}{I_{R}}\times \frac{D_{R}}{D_{S}}\times \frac{n_{S}^{2}}{n_{R}^{2}},$$
where Φ is the PL QY, *I* is the integrated PL intensity, *D* is the absorbance, *n* is the refractive index of the respective solvents, and the subscripts *S* and *R* stood for the sample and reference, respectively.

### 2.5. Detection of Fe^3+^ Cations

The reaction of potassium hexacyanoferrate (II) with Fe^3+^ produces a poorly soluble precipitate: Fe^III^Cl_3_ + K4[Fe^II^(CN)_6_] → KFe^III^[Fe^II^CN_6_] + 3KCl. The resulting potassium-iron (III) hexacyanoferrate (II) is slightly soluble to form a colloidal solution. After incubation of RAW264 macrophage cell line with MLNCs for 20 h, cells were washed with PBS and fixed with PFA. After repeated washing, 4% potassium hexacyanoferrate solution and 4% HCl solution were added to the cells in a 1:1 ratio. The cells were washed with PBS after 20 min of incubation. The colorimetric response of MLNCs to potassium hexacyanoferrate (II) was studied using an upright microscope Axio Imager Z1 with an HRC CCD camera (ZEISS, Oberkochen, Germany).

### 2.6. Cell Culture and Assessment of Cell Cytotoxicity

#### 2.6.1. Cells

Cell lines of human embryonic kidney HEK293, human lymphocytes Jurkat, murine macrophages RAW264, J774, and chondrocytes ADCT5 were provided from the collection of the Shemyakin-Ovchinnikov Institute of Bioorganic Chemistry RAS (Moscow, Russia). Cells were cultivated in DMEM or RPMI-1640 culture medium supplemented by 7% of fetal bovine serum, glutamine, penicillin/streptomycin. Adherent cells were detached using 0.05% trypsin-EDTA (all from PanEco, Moscow, Russia), counted and sub-cultured. 

#### 2.6.2. MTT Assay

The cytotoxicity of MLNCs was investigated by MTT assay [39]. The solutions containing M-CDs were diluted to yield a concentration range from 150 to 1μg/mL. The cells were seeded at 10,000 cells/well and incubated for 72 h in a CO_2_ incubator at 37 °C. MTT was added to each well for the last 4 h. The culture medium was eliminated from the wells, and formazan crystals were dissolved in 100 μL of DMSO for 20 min. Cytotoxic concentration, giving 50% of the maximal toxic effect (IC50), was calculated from the titration curves. The inhibition of proliferation (inhibition index, II) was calculated as II = (1 − ODexperiment)/ODcontrol, where OD was the formazan optical density.

### 2.7. Flow Cytometry

M-CDs or sM-CDs dissolved in RPMI were titrated in 96-U-bottom well plates. Jurkat cells (104/well) were added to the MLNCs and incubated for 24 h. Wells containing medium without MLNCs were used as control. Before the analysis, cells were washed once with saline, and the fluorescence response of cells and the toxicity of M-CDs and sM-CDs were analyzed by a MACSQuant flow cytometer (Miltenyi, Bergisch Gladbach, Germany). The toxicity was estimated by propidium iodide inclusion.

### 2.8. Efficiency of M-CDs Accumulation in the Cells

HEK293 and Jurkat cells were cultivated with M-CDs or sM-CDs in 96-flat bottom cell culture plates, as described above. After overnight incubation, the supernatants were removed and the fluorescent response in supernatants was measured at excitation at 365 nm and recorded in the wavelength range of 410–460 nm using a spectrofluorometer GlomaxMulti (Promega, Fitchburg, MA, USA). The accumulation efficiency was determined as follows: the percentage of M-CDs entering the cells was estimated by the formula (3):(3)$$V=\left(1-\frac{I}{I_{0}}\right)\times 100\%,$$
where *V*—percentage of M-CDs or sM-CDs in cells, *I*—PL intensity of the supernatant, *I*_0_—PL intensity of the control group M-CDs or sM-CDs without cells.

### 2.9. Confocal Fluorescence Imaging

HEK293 cells (105) were grown on sterile cover slides placed in 6-well plates. After the adhesion of the cells overnight, the cells were incubated with various concentrations of MLNCs for 24 h. Then, the cells were stained with membrane tracker WGA-Alexa555 and Nuclear Green fluorescent dyes (Molecular Probe, Eugene, USA). The cells were then washed, fixed with 4% paraformaldehyde for about 15 min, washed again, and polymerized with Mowiol 4.88 medium (Calbiochem, Nottingham, UK). Hoechst 33,342 (Sigma, USA) was used to visualize nuclei. Slides were analyzed using an Eclipse TE2000 confocal microscope (Nikon, Tokyo, Japan) at a wavelength of 405 nm.

### 2.10. Statistics

Graphs were created using MS Excel software. The data are represented as mean ± standard error of mean (SEM) of at least three independent experiments or as one representative experiment from three. Statistical analysis was performed using Student’s t-test. Significance levels of *p* < 0.05 were considered statistically reliable.

## 3. Results

### 3.1. Characterization of the Morphology and Optical Properties of M-CDs

The size and morphology of the M-CDs and sM-CDs were characterized using scanning electron microscopy (SEM) in STEM mode. Both M-CDs and sM-CDs demonstrated quite narrow size dispersion (Figure 1). The darker areas represent the iron component of M-CDs (Figure 1a, insert, black arrows) within the carbon shell, which is slightly lighter in the shade of gray since CDs have a lower electron density compared to the iron component. Contrary to it, sM-CDs have a uniform gray color (Figure 1b, inset) as it is not possible to distinguish between darker impregnations of the iron component. This could be associated with the presence of the glutathione stabilizer on the sM-CDs surface.

Based on the SEM images, the geometric sizes were determined. The average size of the M-CDs and sM-CDs were 33 ± 6 and 61 ± 14 nm, respectively (Figure 1c,d). The size difference can be explained by the stabilization which was carried out before the precipitation of M-CDs. Possibly larger particles were retained in the sM-CDs solution. Iron was detected in the EDX spectra, which may also indicate the successful formation of MNPs in the carbon matrix of M-CDs and sM-CDs (Appendix A). In addition, the analysis of the EDX statistics of M-CDs and sM-CDs did not show significant changes in the composition after stabilization (Appendix A).

DLS data showed a narrow size distribution with an average hydrodynamic diameter of the M-CDs and sM-CDs in water of about 65 and 120 nm, respectively (Figure 1e). Zeta potentials (ζ) of the M-CDs and sM-CDs in water were 0 and −20 mV, respectively. Glutathione charges the M-CDs surface, preventing sM-CDs aggregation due to the electrostatic repulsion. These results demonstrate that the surface of sM-CDs exposes negatively charged hydrophilic groups, which contribute to the dispersibility and stability of sM-CDs in water. The hydrodynamic diameter of MLNCs was two times larger in a comparison with the physical size obtained by the processing of SEM images. This can be explained by the presence of a solvation shell around the particles due to the effect of hydration [19].

The optical properties of M-CDs and sM-CDs were studied using UV-visible absorption and PL spectroscopy. The absorption spectra exhibit peaks at a wavelength of 350 nm and a low-intensity absorption shoulder in the wavelength range of 400–500 nm (Figure 2, black curves), which is typical for CDs [36]. At the same time, the contribution of MNPs manifests itself in a monotonic increase in the absorption to the short-wavelength region, which is typical for the absorption spectra of Fe_3_O_4_ [40]. Thus, the electronic absorption spectra of MLNCs demonstrate both the features of CDs and MNPs. M-CD and sM-CD are characterized by the same shape of absorption spectra, except for a small difference in higher optical density in the 400–600 nm arm wavelengths of sM-CD compared to M-CD with the same optical characteristics peak density, which can be explained by more presence of the magnetic component.

The PL spectra of MLNCs (Figure 2a,b) exhibit intense emission bands at a wavelength of 450 nm at 350 nm excitation. Aqueous solutions of M-CDs and sM-CDs have an orange color in daylight and a bright blue-green PL emission with UV irradiation at 365 nm (Figure 2c,d). In addition, MLNCs exhibit excitation-dependent emission, when, with increasing excitation wavelength, a long-wavelength shift of the emission maximum with a decrease in the PL intensity, which is characteristic of the PL emission from carbon dots [41]. This behavior of PL, similar to CDs, is also a confirmation that the luminescent part of the composites is responsible for the emission of MLNCs.

The study of the photoluminescence excitation spectra of MLNCs (Figure 2a,b, magenta lines) showed that for the long-wavelength radiation, the luminescence shoulder with a maximum of 550 nm is provided by states absorbing not only at 340 nm in accordance with the main absorption peak but also secondary peaks for both types of composites. In the case of M-CDs, there are two additional peaks at 450 and 515 nm, while for sM-CDs there is only one additional peak at 450 nm. This indicates that the sample had several absorption states leading to long-wavelength radiation, which is also characteristic of CDs [41].

A photograph of an aqueous solution of M-CDs before (Figure 2c) and after stabilization (Figure 2d) under a UV lamp in the presence of an external magnet demonstrates the magnetic properties of M-CDs and sM-CDs. Herein, it can be seen that sM-CDs have bright photoluminescence near the magnet, while M-CDs are colored over the entire volume of the solution and a gradient of PL intensity is observed near the magnet. Thus, this is evidence of the stronger magnetic properties of sM-CDs.

The PL lifetime of MLNCs was measured using a time-resolved fluorescence microscope. The luminescence decay curve is very well fitted to a double-exponential function, as shown in Figure 3a,b. The PL decay curves of stock M-CDs (red) and stabilized sM-CDs (black) in water and DMEM are shown in Figure 3a,b, respectively.

The average PL lifetime of stock M-CDs and sM-CDs is 10 ns and 11 ns, respectively (Appendix A). Such a short PL lifetime is indicative of the radiative recombination nature of the excitations. As can be seen, the difference in the PL lifetimes of M-CDs is extremely small, which may indicate an insignificant effect of stabilization on the PL decay times. Moreover, the same PL decay kinetics of nanocomposites before and after stabilization indicates that the PL QY does not change. In order to further investigate the PL properties of M-CDs, the PL QY, and the effect of the external factors on the PL were investigated. PL QY of the M-CDs and sM-CDs were about 20% and 28%, respectively.

### 3.2. Investigation of M-CDs Photostability

Stability is one of the key parameters when using nanocomposites to study biological processes in cells, since decomposition can be observed in living organisms during the prolonged circulation of nanocomposites in the bloodstream. Therefore, the photostability of the M-CDs and sM-CDs was further investigated. To carry out this, MLNCs were dissolved in DMEM and then exposed to irradiation of light with a wavelength of 365 nm for 20 min. The exposure was calculated using the following formula:(4)A=P×t/S,
where *A*—the exposure, *P*—the radiation power, *t*—the exposure time, *S*—the absorption area of the radiation.

Then the exposure is equal to:(5)$$A=3\;\mathrm{mW}\times 20\times 60\;s/1\;{\mathrm{cm}}^{2}=3.6\;W\cdot s\cdot {\mathrm{cm}}^{-2}$$

The PL intensity of the MLNCs will remain fairly stable at an irradiation exposure of 3.5 W·s·cm^−2^ (Figure 3c). At the same time, sM-CDs (black line) demonstrate a smaller decrease in the PL intensity, which indicates their higher stability compared to M-CDs (red line). The photostability of the PL intensity of the composites was studied at different excitation waves since different luminescent centers in both M-CDs and sM-CDs are responsible for the emission of the composites. The obtained dependences of the PL intensity on the exposure differ when changing the excitation wavelength (Figure 3c,d). This is due to the luminescent nature of the composites, which are similar to CDs. In this case, luminescent centers that emit at 350 nm excitation absorb more irradiation energy than those that emit at 405 nm; this causes a difference in the dependences of the stability of nanocomposites. Thus, M-CDs and sM-CDs demonstrated good photostability in culture medium DMEM.

Hence, it can be seen that the obtained MLNCs have a size of the order of 30–60 nm and have good solubility in water. In addition, M-CDs and sM-CDs exhibit strong magnetic properties and attractive optical properties, such as absorption in a wide spectral range, high QY of PL, PL lifetimes of 10 ns order, and good photostability. Thus, the obtained M-CDs and sM-CDs have the potential for bioapplication.

### 3.3. Efficiency of M-CDs Accumulation in Cells

The efficiency of M-CDs accumulation in cells was investigated by using the spectrofluorometer. HEK293 (adhesive), RAW264 (semi-adhesive), and Jurkat (suspension) cells were incubated with different concentrations of M-CDs or sM-CDs for 48 h. Wells containing M-CDs without cells were used as controls. After cell incubation, supernatants from cell-containing wells were removed; cells were washed once with PBS, 200 uL of fresh PBS were added and fluorescence was analyzed using the spectrofluorimeter. The PL signal in the cells was calculated as a percentage of the total amount in control cell-free cultures. Maximal uptake was found in lymphoid Jurkat cells and minimal in epithelial HEK293 cells both for M-CDs and sM-CDs (Figure 4a,b). Particle uptake was dose-dependent and reached 8% and 12% at the maximal M-CDs and sM-CDs particle concentration, accordingly. There was a plateau of uptake at high NPs concentrations.

### 3.4. Cytotoxicity of M-CDs

Analysis of cell proliferation of ATDC5 (epithelial cells), RAW264 (macrophages), and Jurkat (lymphocytes) cells in the presence of different concentrations of NPs demonstrated average cytotoxicity at high NP concentrations (Figure 4c,d). Higher toxicity of both M-CDs and sM-CDs was found against lymphoid cells (Figure 4c,d, blue curves). On average, 20% toxicity is considered compatible with bioapplications. Low doses of the particles stimulated epithelial cell proliferation (index of cytotoxicity < 0), supporting the biocompatibility of MLNCs for healthy tissues. Furthermore, sM-CDs were slightly more toxic at high concentrations than M-CDs (II = 0.38 vs. 0.2). 

As an alternative method, a flow cytometry analysis was used to assess the cytotoxicity of M-CDs and sM-CDs estimated by propidium iodide cell staining. Both M-CDs and sM-CDs were nontoxic and stimulated cell survival and proliferation at low doses (Figure 5a,b). Straight lines of the corresponding colors show the background level of cell death in control wells. The results were reliable and reproduced in two different cell lines. Representative dot plots show slightly higher cytotoxicity of sM-CDs (Figure 5e,f), which supports the results of the MTT assay (Figure 4c,d). The total PL signal was higher in M-CDs (Figure 5c). This can be a result of NPs size, cell capacity to accumulate the particles, and brighter fluorescence of unshelled M-CDs, as is seen in the histograms (Figure 5d, blue).

Both dot plots and histograms show that all cells are stained by MLNCs either intracellular or via the surface binding and retention of NPs (Figure 5d–g). These results indicate excellent biocompatibility of MLNCs in a broad concentration range. Thus, it can be concluded that M-CDs are cytocompatible, which is important for bioimaging applications.

### 3.5. Confocal Fluorescence Imaging

The results obtained by flow cytometry and spectrofluorimetry cannot separate the extracellular localization of MLNCs from intracellular ones. To understand their distribution in the cells, confocal microscopy was used. Adhesive HEK293 cells were grown on cover slides in the presence of M-CDs and sM-CDs for 20 h, fixed and permeabilized. The particle PL is visible under 408 nm laser excitation (Figure 6, blue). To visualize cell membranes, WGA-Alexa555 was used (Figure 6, red). To localize the nuclear accumulation of NPs, cells were stained with Nuclear Green fluorescent dye (Figure 6. green). The results demonstrated that smaller M-CDs, but not larger sM-CDs, accumulated mostly in the nuclei of the cells (Figure 6a–c, white arrows), and more accurately—in the nucleoli (Figure 6d–f, ovals and red arrows). Nucleoli are areas of chromosomes on which ribosomal ribonucleic acids are synthesized. These places with less condensed DNA leave space for NPs accommodation.

Larger sM-CDs were found only in the cytoplasm of the cells, as evidenced by empty nuclei (Figure 6g–i, yellow arrows).

### 3.6. Detection of Fe^3+^ Cations

Potassium hexacyanoferrate (II) is used in analytical chemistry as a reagent for the detection of certain cations. For example, a poorly soluble blue precipitate is formed in the reaction with Fe^3+^. Thus, by studying the colorimetric response of MLNCs to potassium hexacyanoferrate (II), it is possible to determine the presence of a magnetic component. After incubation of semi-adhesive RAW264 cells with MLNCs for 20 h, a solution of potassium hexacyanoferrate was added to the cells. Cell images from the optical microscope show a blue color, which indicates a successful reaction of potassium hexacyanoferrate with the ferric iron (Figure 7). Cells incubated with sM-CDs have a brighter color than with M-CD (Figure 7a,b). Consequently, upon penetration into cells, sM-CDs nanocomposites have a larger iron part than M-CDs. At the same time, the stabilization of nanocomposites prevents the oxidation of ferric iron to ferrous and prevents the decomposition of M-CDs upon accumulation by cells. Thus, the stronger blue coloration of sM-CDs indicates the integrity of the magnetic component in the nanocomposites.

## 4. Discussion

We report a one-step synthesis of an MLNCs surface stabilized for potential biomedical applications. The morphology, optical properties, and elemental analysis of the obtained MLNCs were characterized by SEM, EDX, DLS, UV-Vis, and time-resolved spectroscopy. The stabilization of MLNCs obtained by the hydrothermal synthesis method improved their optical, physical, and chemical properties, which distinguishes them from similar composites obtained by other research groups [22,24,42,43].

Several mechanisms for the formation of composites based on precursors of carbon dots and MNP are known [44,45]. Here, the hydrothermally synthesized magnetic cores are used as the crystal growth seeds for the formation of carbon dots due to the coordination of Fe^2+^/Fe^3+^ ions with the carbonyl groups of carbon dots in accordance with [13]. 

The size of the obtained nanocomposites varies from 30 to 60 nm depending on the stabilization, which is an order of magnitude smaller than other nanocomposites (about 600 nm) obtained by LBL methods with sacrificial templates [17,46,47].

It was established that MLNCs exhibit the luminescence properties of traditional CDs, namely, the position and width of the PL bands, as well as the dependence on the wavelength of the exciting light [36,41]. The luminescence behavior of MLNC does not change as a result of stabilization, while the PL QY increases from 20% to 28%. The PL QY obtained after stabilization exceeds the same parameter as other similar systems obtained earlier [19,48]. An analysis of the PL decay kinetics showed that the average PL lifetimes of MLNCs almost do not change before and after stabilization, regardless of the aqueous or DMEM solvent, and is about 11 ns, which is longer than the lifetimes of similar systems obtained in [20]. This fact indicates that the presence of a magnetic component in the composition of MLNC does not significantly affect the PL QY of the luminescent part of CDs. The observed effect is due to the formation of a carbon matrix with embedded MNPs, where the luminescent centers are located at a sufficient distance to avoid energy/charge transfer. This special luminescent nature is one of the main advantages of CDs when combined into one MLNC with a magnetic component in comparison with traditional semiconductor QDs (CdSe, CIS, etc). This effect is observed in the paper [49], when the creation of a nanocomposite of CdSe/ZnS quantum dots and Fe_2_O_3_ magnetic NPs results in the quenching of the QD PL three times.

Magneto-luminescent systems obtained by a similar synthesis, but without stabilization do not demonstrate colloidal stability [24,50]. However, in addition to the better photostability of sM-CD compared to M-CD, stabilization also prevented the aggregation of MLNCs, and colloidally stable solutions were obtained. The carbon dot core can be covered by a variety of surface functional groups, such as carbonyl, carboxyl, amines, or thiols. The use of high temperature/high-pressure treatment may trigger many simultaneous reactions, resulting in the most reactive functionalities of the precursors on the carbon dot after the synthesis [51]. We assume that the post-treatment of the M-CDs obtained using glutathione leads to the colloidal stabilization of the nanostructures due to the chemical interaction of functional groups of glutathione (by –COOH or/and –NH_2_) to the variety of the surface functional groups on the carbon dot core. The stretched-out molecules of the attached glutathione are responsible for the electrostatic repulsion between the M-CDs.

Cytotoxicity results show that MLNCs have excellent biocompatibility, even at sufficiently high concentrations, which is the main advantage for bioapplications. In the same way in paper [19], an in vitro study of the toxicity of superparamagnetic nitrogen-doped carbon-iron oxide hybrid QDs (C-Fe_3_O_4_ QDs) was carried out. Human cervical carcinoma cells (HeLa) were incubated with C-Fe_3_O_4_ concentrations of up to half of 500 μg/mL for 24 and 48 h. The results of the analysis showed that C-Fe_3_O_4_ QDs do not exhibit significant cytotoxicity, while the viability of cells incubated with similar carbon nanodots doped superparamagnetic iron oxide NPs (FeCD) also showed no cytotoxicity in the concentration range up to 20 μg/mL [22], which is several times less than the concentration used in our work.

We have demonstrated that MLNCs are predominantly localized in the nuclei of cells, while the stabilized ones do not demonstrate such an effect. Different localization may be associated either with an increase in the size of MLNCs after stabilization (as nucleus membrane pores are around 30–50 nm) or the charge of MLNCs. The latter is more possible, as Liu et al. demonstrated that after incubation with 50, 100 or 150 μg/mL QD C-Fe_3_O_4_ for 24 h, the PL signal was observed in the cytoplasm, but not in the nucleus [19]. In another article [22], the localization of FeCD was also limited to the cytoplasm, since a fluorescent signal was observed in the cytoplasm, leaving the nucleus as a dark area aside. Analysis of the ferric test confirmed that stabilization of the MLNC surface prevents oxidation of ferrite, which helps to maintain magnetic properties.

Thus, since MLNCs exhibit small size, high stability, and efficient magnetic and optical response, as well as high biocompatibility and intracellular localization, the proposed one-step synthesis of MLNCs with surface stabilization has a potential for biomedical applications.

## 5. Conclusions

In summary, we have synthesized MLNC in one stage from the precursors of CDs and MNPs. The reliable MLNCs resistant to breakdown were obtained after surface stabilization. The optical properties of the MLNCs were characterized, and comparative analysis and the effect of stabilization were performed. In addition, cytotoxicity, intracellular localization, and the luminescent response of cells with MLNCs were studied.

Thus, based on the above features, MLNCs combining luminescent and magnetic properties could be exploited for theranostics in biological applications, such as bioimaging and hyperthermia in a magnetic field.

## Figures and Tables

**Figure 1 nanomaterials-12-01396-f001:**
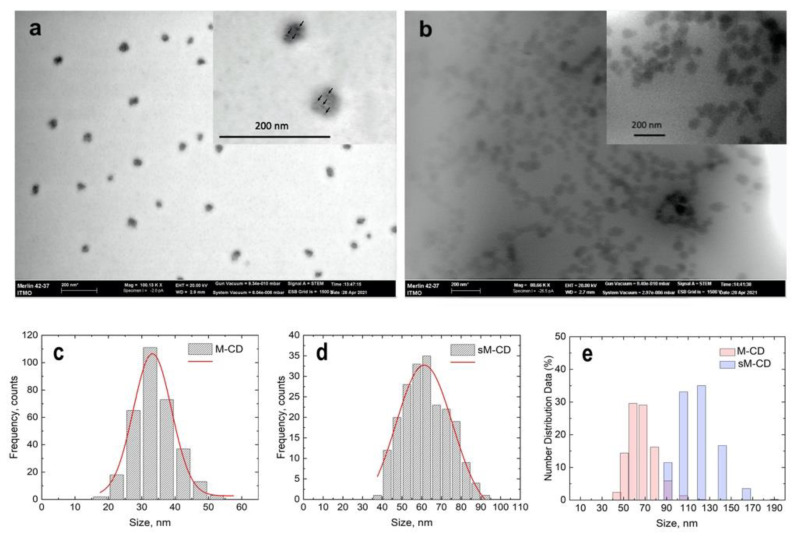
Characteristics of MLNCs. STEM images of the stock M-CD (**a**) and sM-CDs (**b**). Insets show a higher magnification. Black arrows show iron component of M-CDs (**a**, insert). Scale bar corresponds to 200 nm. Dry (**c**,**d**) and hydrated (**e**) diameters of M-CDs and sM-CDs.

**Figure 2 nanomaterials-12-01396-f002:**
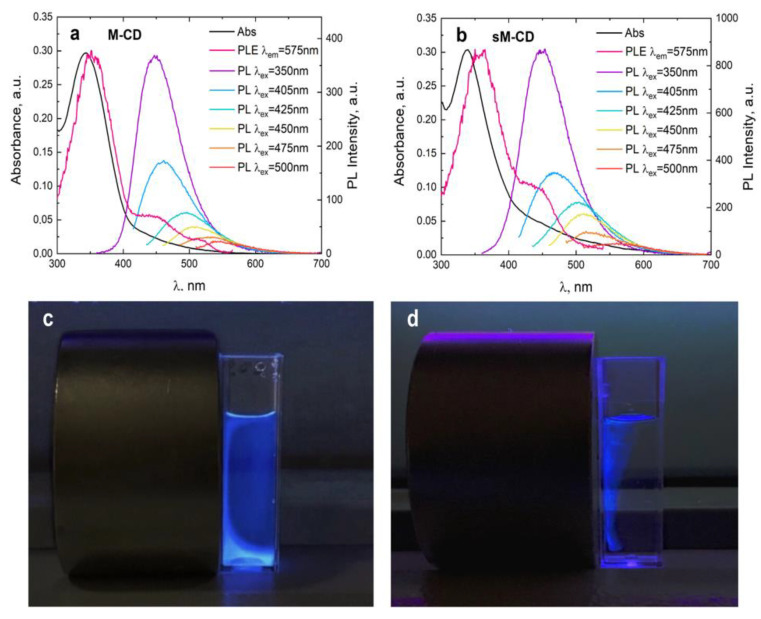
Characteristics of MLNCs. Absorption, PL excitation, PL spectra upon different wavelengths excitation and overview near and external magnet of M-CDs (**a**,**c**) and sM-CDs (**b**,**d**).

**Figure 3 nanomaterials-12-01396-f003:**
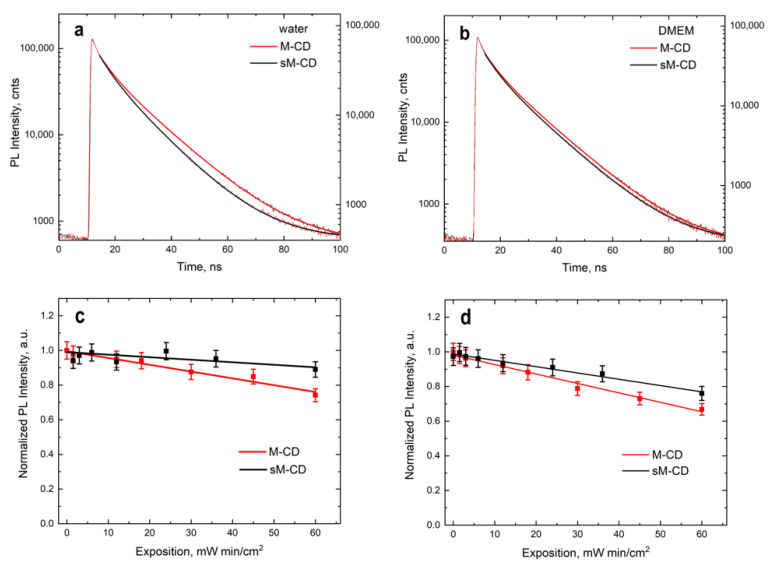
Characteristics of MLNCs. PL decay curves in water (**a**) and in DMEM medium (**b**) upon excitation at 405 nm and intensity upon excitation of 350 nm (**c**) and 405 nm (**d**) of M-CDs (red) and sM-CDs (black) on the time of exposure to UV radiation.

**Figure 4 nanomaterials-12-01396-f004:**
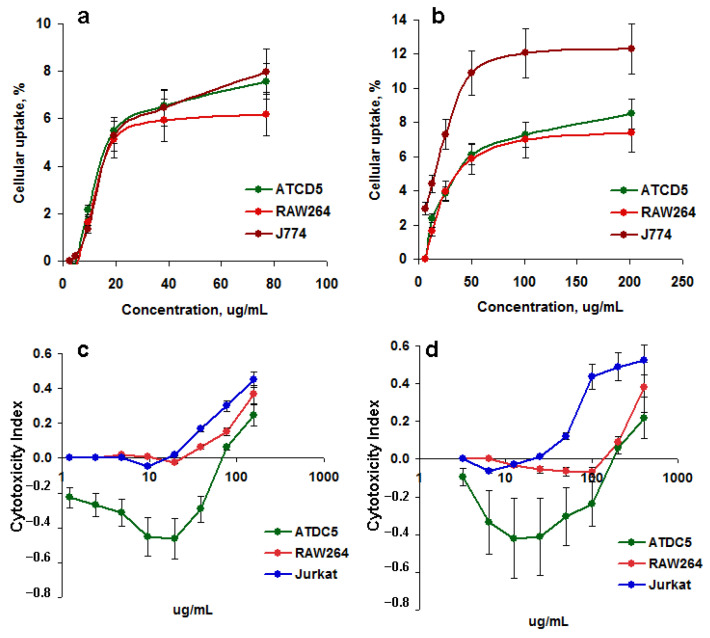
MLNCs interaction with live cells. Cellular uptake by (**a**,**b**) and cytotoxicity (**c**,**d**) of M-CDs (**a**,**c**) and sM-CDs (**b**,**d**) to the cells. Epithelial (ATCD5) and macrophage (RAW264, J774), cells were incubated for 48 h (**a**,**b**) and cellular uptake was analyzed. ATDC5, RAW264, and Jurkat cells were incubated for 72 h and cytotoxicity were analyzed by MTT.

**Figure 5 nanomaterials-12-01396-f005:**
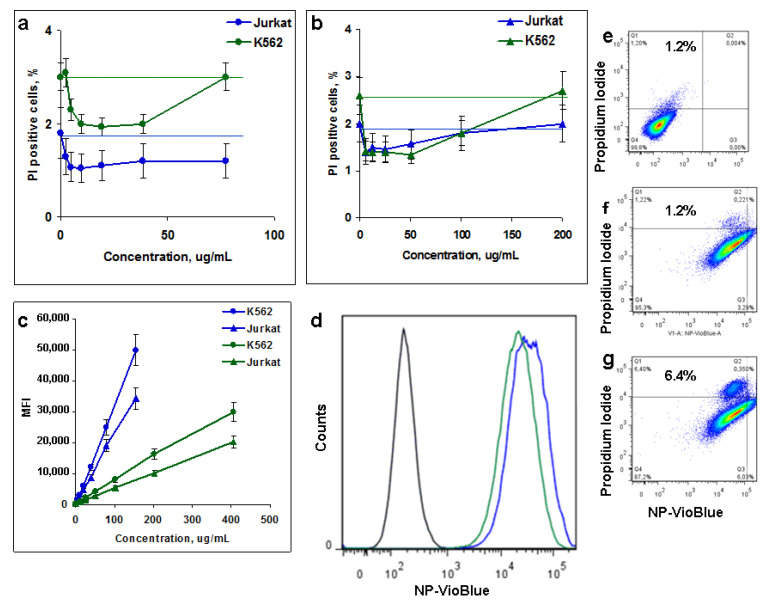
Flow cytometry characteristics of M-CDs particles. (**a**,**b**): Percentage of K562 and Jurkat propidium iodide positive cells incubated for 48 h with M-CDs (**a**) and sM-CDs (**b**,**c**): Dose-dependent K562 and Jurkat cell staining by M-CDs (blue) and sM-CDs (green). **d**–**g**: Histograms (**d**) and representative dot plots (**e**–**g**) of unlabeled (**d**, gray, **e**) or M-CDs (**d**, blue, **f**) and sM-CDs (**d**, green, **g**) K562 cells. **e**–**g**: NPs fluorescence corresponds to VioBlue channel (abscissa axis) and propidium iodide—to dead cells (ordinate axis).

**Figure 6 nanomaterials-12-01396-f006:**
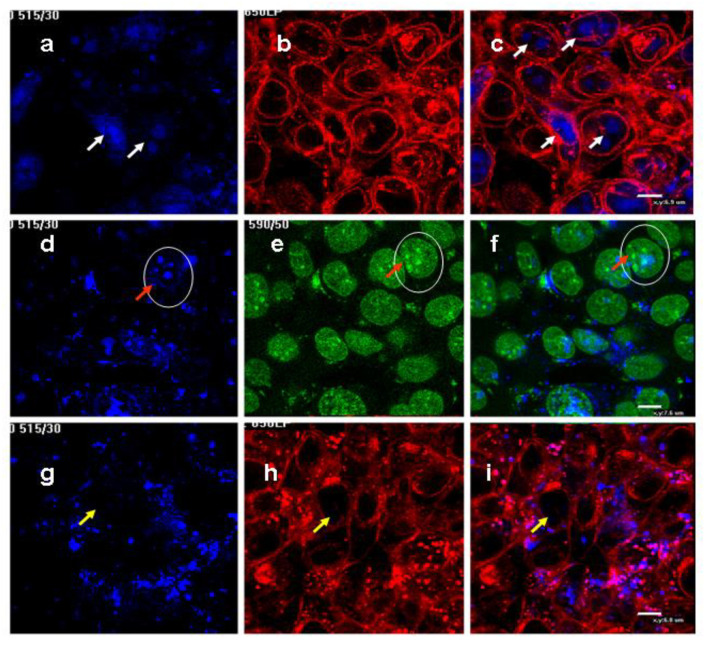
Confocal images of HEK293 cells cultivated in the presence of M-CDs (**a**–**f**) or sM-CDs (**g**–**i**). Cell and nuclei membranes are stained in red and nuclei in green. Scale bar 6–7 μm. White arrows show nuclear localization of M-CDs (**a**,**c**); red arrows show nucleoli localization of M-CDs (ovals and red arrows); yellow arrows show the absence of the sM-CDs in the cell nuclei.

**Figure 7 nanomaterials-12-01396-f007:**
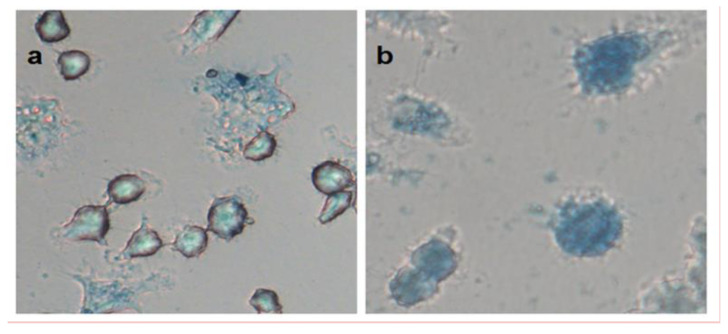
Images of macrophage cells RAW264 incubated with M-CDs (**a**) and sM-CDs (**b**) in the presence of potassium hexacyanoferrate (II).

## Data Availability

Not applicable.

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
