# Peer review of "Magneto-Luminescent Nanocomposites Based on Carbon Dots and Ferrite with Potential for Bioapplication"

_nanomaterials, 2022, doi:10.3390/nano12091396_

Round 1

Reviewer 1 Report

The article presents on multifunctional nanocomposites that combine both magnetic and photoluminescent (PL) properties provide for nanomedical applications. The authors have developed a one-stage microwave synthesis of magnetoluminescent -magnetic carbon dots (M-CDs) nanocomposites with subsequent stabilization with L-glutathione.

The resulting MLNCs demonstrated good dispersibility in water, tunable PL with a quantum yield of up to 28%, high photostability, and good cytocompatibility, showing M-CDs localization in the cell nuclei.

The study is interesting and nicely performed and presented.

There are some parts that should be improved before the manuscript can be considered complete for acceptance.

  1. In the synthesis part the temperature in the microwave-assisted process has to be specified. The power and duration of the reaction does not provide sufficient details. As the process is most likely hydrothermal, the pressure built in the system should also be given,as the microwave system utilized in the work has pressure feedback.
  2. The term “soluble/dissolved” is used in the manuscript in several places, referring to the dispersion of the nanoparticles in water or DMIM. This term should more properly be changed to “dispersible” as the author already used in the abstract.
  3. As for the nanoparticle composition no data is presented. Table S1 present the statistic of elemental compositions, where it is not clear if the data is on the atomic percent or weight percent. The type of the data has to be specified in the captions of Table S1 and S2.
  4. XRD is recommended to be performed to reveal the composition of the magnetic nanoparticles.
  5. Few lines on the formation mechanism of the carbon around the magnetic core should be given.
  6. Molecular formula should be given with the proper use of subscripts.
  7. Line 240; the word “salvation” should change to “solvation”
  8. References should be numbered

Author Response

Dear Reviewer,

We are very thankful to our reviewers. Please see the attachment.

Reviewer 2 Report

The article titled ‘Magneto-luminescent nanocomposites based on carbon dots and ferrite for efficient bioimaging with the possibility of hyperthermia’ pertains to the one-pot synthesis of luminescent carbon dots and magnetic iron oxide nanoparticles and their possible application in bio-imaging. The magneto-luminescent nanocrystals (MLNCs) have been characterized using various spectroscopic and microscopic techniques. The MLNCs have been subjected to different cytotoxic studies involving different cell lines as well as confocal studies. The synthetic method as well as the material obtained seems to be novel but it is hard to believe. The authors have indicated three-step, two-step and one-step synthetic methods, and they suggest that all three methods lead to the same materials, but they do not give sufficient arguments to support this and it is hard to believe. Yet, the article fails to convincingly convey the advantage of having a nanocrystal with the dual property of luminescence and magnetism. The authors have not explored the magnetic properties of the materials sufficiently enough to claim a possible use of these nanomaterials for hyperthermia. The authors claim to have synthesized stable M-CDs by post-treatment with glutathione yet the zeta potential value was -20mV which is still considered to be unstable as a stable colloidal zeta potential value is |30|mV and above. The authors were also unclear in the manuscript as to which of the nanocrystals they would favour for bio-imaging as the sM-CDs were cytotoxic and localized in the cytoplasm in comparison to M-CDs which localized in the nuclei but had lower colloidal stability. Nevertheless, it is a good attempt at synthesising such hybrid nanomaterials but would require further clarifications and studies.

  1. In Fig. 1(a, b) as well as in the manuscript the authors mention using SEM for determining the size of the MLNCs but the figure seems to be transmission electron microscope (TEM) images. Would the authors check and clarify it?
  2. Why increased stability of M-CDs is obtained by using glutathione. Justify the nature of the interaction between glutathione and M-CDs and why glutathione provides increased stability. Perhaps, an IR of these particles could show the surface groups.
  3. For the synthesis, all the precursors are added together and autoclaved. Explain why the precursors would form such a hybrid material rather than forming individual magnetic or luminescent properties. Also, if such individual particles are formed, they would be smaller in size compared to the hybrid material meaning they would be suspended in the supernatant after centrifugation but the authors have used the supernatant to obtain the particles. Why?
  4. In Fig. 4 (a, b) describing the accumulation of MLNCs in different cell lines, at higher concentrations the authors claim the cellular uptake to plateau. In order to see the saturation of cellular uptake, one more measurement at the least is needed, particularly for M-CDs in RAW264 and Jurkat as a decrease rather than a plateau is seen at the endpoint.
  5. The authors claim the M-CDs have better localization in the nuclei to be smaller sized in comparison to sM-CDs. Why not consider the charge on these particles as M-CDs is neutral while the other is negatively charged? It is also interesting to see negatively charged sM-CDs in the cytoplasm considering it has to cross a negatively charged cell membrane.
  6. As the article deals with exploring the properties of both luminescent and magnetic nanoparticles, it would be better to perform some magnetic measurements such as SQUID to obtain the magnetization values and the hysteresis curve.
  7. Regarding the writing,

-The authors have to take care using proper notations such as Fe3O4 (line 63,247, 450), sp2-hybridization (line 74) among many others.

-Introduction may need further checking for grammatical errors and framing such as in lines 37 and 77.

-Fig. 5d describing flow cytometry mentions the graph as histograms but are rather Gaussian curves.

Author Response

(The authors gave the same response as above.)

Round 2

Reviewer 2 Report

The authors have considered the comments and improved the article significantly. They have been able to address most of the queries. The authors have further explained the formation of such heterostructures including relevant references and the chemical interactions involved in stabilizing the particles. The explanations seem adequate, at the moment, but they have not convinced me.
Even though further studies are required to explore the magnetic properties of the material, the paper in the present form seems to be acceptable.

Specific comments:
1. Yes, as the authors have suggested it would be better to change the fig. 4 (a,b) with the ones given in the cover letter as they seem more accurate.
2. In the abstract, in line 21, the sentence requires reframing.

Author Response

(The authors gave the same response as above.)
